# Gain- and Loss-of-Function *CFTR* Alleles Are Associated with COVID-19 Clinical Outcomes

**DOI:** 10.3390/cells11244096

**Published:** 2022-12-16

**Authors:** Margherita Baldassarri, Kristina Zguro, Valeria Tomati, Cristina Pastorino, Francesca Fava, Susanna Croci, Mirella Bruttini, Nicola Picchiotti, Simone Furini, Nicoletta Pedemonte, Chiara Gabbi, Alessandra Renieri, Chiara Fallerini

**Affiliations:** 1Medical Genetics, University of Siena, 53100 Siena, Italy; 2Med Biotech Hub and Competence Center, Department of Medical Biotechnologies, University of Siena, 53100 Siena, Italy; 3UOC Genetica Medica, IRCCS Istituto Giannina Gaslini, 16148 Genova, Italy; 4Department of Neurosciences, Rehabilitation, Ophthalmology, Genetics, Maternal and Child Health (DINOGMI), University of Genoa, 16126 Genoa, Italy; 5Genetica Medica, Azienda Ospedaliero-Universitaria Senese, 53100 Siena, Italy; 6Department of Mathematics, University of Pavia, 27100 Pavia, Italy; 7University of Siena, DIISM-SAILAB, 53100 Siena, Italy; 8Department of Biosciences and Nutrition, Karolinska Institutet, 17177 Stockholm, Sweden

**Keywords:** COVID-19, *CFTR* complex alleles, post-Mendelian model

## Abstract

Carriers of single pathogenic variants of the *CFTR* (cystic fibrosis transmembrane conductance regulator) gene have a higher risk of severe COVID-19 and 14-day death. The machine learning post-Mendelian model pinpointed *CFTR* as a bidirectional modulator of COVID-19 outcomes. Here, we demonstrate that the rare complex allele [G576V;R668C] is associated with a milder disease via a gain-of-function mechanism. Conversely, *CFTR* ultra-rare alleles with reduced function are associated with disease severity either alone (dominant disorder) or with another hypomorphic allele in the second chromosome (recessive disorder) with a global residual CFTR activity between 50 to 91%. Furthermore, we characterized novel *CFTR* complex alleles, including [A238V;F508del], [R74W;D1270N;V201M], [I1027T;F508del], [I506V;D1168G], and simple alleles, including R347C, F1052V, Y625N, I328V, K68E, A309D, A252T, G542*, V562I, R1066H, I506V, I807M, which lead to a reduced CFTR function and thus, to more severe COVID-19. In conclusion, *CFTR* genetic analysis is an important tool in identifying patients at risk of severe COVID-19.

## 1. Introduction

Since the global pandemic of coronavirus disease 2019 (COVID-19), caused by SARS-CoV-2, began in 2020, one of the major challenges has been the identification of the main prognostic factors determining the clinical outcome in order to implement effective preventive and therapeutic strategies. 

It has already been clearly demonstrated that age, sex, and the presence of other pre-existing disorders, such as diabetes or cardiovascular disease, influence the different prognoses of some patients [1,2,3]. It is now well recognized that host genetic factors also play a crucial role in determining COVID-19 clinical outcomes and so far, various common [4,5,6,7,8,9,10,11] and rare [12,13,14,15,16,17] genetic variants are already described in association with susceptibility and severity. In particular, our GEN-COVID consortium has previously shown that the individuals carrying single pathogenic variants of the *CFTR* gene, i.e., CF-carriers, were more likely to undergo severe COVID-19, with a higher risk of 14-day mortality [17] and their geographical distribution correlated with COVID-19 spread and fatalities in 37 countries [18]. Moreover, the application of the new machine learning post-Mendelian model to a large international cohort of COVID-19 patients extracted the *CFTR* gene as an important factor involved in the modulation of COVID-19 outcomes [19,20]. CFTR is a chloride and bicarbonate channel expressed mainly in the lung, liver, pancreas, and intestine. When both *CFTR* alleles are mutated, and the global CFTR activity is impaired by more than 70% (and often more than 95%), (see the CFTR2 database at https://cftr2.org, accessed on 3 November 2022), patients are affected by cystic fibrosis (CF), CF is multiorgan disease, mainly characterized by the high viscosity of secreted fluids, causing plugs and obstructions, and an excessive host inflammatory response [21].

It was recently shown in a preprint that the spike protein of the coronavirus binds to CFTR, inhibiting its activity [22], and the nucleocapsid (N) protein could interact with Smad3, which downregulated CFTR expression [23]. Moreover, WT mice infected with SARS-CoV-2 demonstrated reduced CFTR expression in their lungs [24]. Therefore, we hypothesized that the overall activity of CFTR, determined mainly by the genetic variants that each individual carries, could be a prognostic factor influencing COVID-19 severity. After performing a whole sequencing of the *CFTR* gene in 2,585 individuals and assessing in vitro the CFTR activity related to the identified *CFTR* variants, we show that the balance from COVID-19 mildness to severity is associated with the global CFTR driving force, with a range of global CFTR activity from 110 to 50%.

## 2. Material and Methods

### 2.1. Patients

A cohort of 2585 SARS-CoV-2 PCR-confirmed positive subjects enrolled within the GEN-COVID multicenter study (NCT04549831, https://sites.google.com/dbm.unisi.it/gen-covid, accessed on 3 November 2022) [25] was used in this study. The patients were recruited from March 2021 to March 2022 in 40 hospitals and primary care centers in Italy. All SARS-CoV-2 infections were assumed to be due to the alpha (B.1.1.7) and delta (B.1.617.2) variants, considering their prevalence in Italy during the study period (https://www.gisaid.org/, accessed on 3 November 2022). All the enrolled subjects were adults (aged ≥ 18 years), and either they or their legally authorized representatives provided informed consent for participation. The GEN-COVID study was approved by the University Hospital (Azienda ospedaliero-universitaria Senese) ethical review board, Siena, Italy (Prot n. 16917, dated 16 March 2020) and the local IRBs of all the recruiting hospitals involved. The COVID-19 clinical severity was assessed as previously described [25] using a modified version of the WHO COVID-19 outcome scale [26]. The following 6 categories of severity were identified: (1) death; (2) hospitalized, receiving invasive mechanical ventilation; (3) hospitalized, receiving continuous positive airway pressure (CPAP) or bilevel positive airway pressure (BiPAP) ventilation; (4) hospitalized, receiving low-flow supplemental oxygen; (5) hospitalized, not receiving supplemental oxygen; and (6) not hospitalized. During the follow-up, a questionnaire about the COVID-19 effects in the medium and long term was administered to the patients. “Long-COVID” was defined as the self-reported presence of “any symptoms that cannot be explained by alternative diagnoses, or impact on everyday functioning, at least 3 months after the onset of COVID-19”.

### 2.2. Whole Exome Sequencing Analysis (WES)

WES with at least 97% coverage at 20× was performed using the NovaSeq6000 System (Illumina, San Diego, CA, USA) as previously described [25]. The WES data were represented in a binary mode on a gene-by-gene basis [19,20].

### 2.3. Chemicals, Vectors, and Antibodies

FRT and CFBE41o- cells were transiently transfected with vectors carrying specific *CFTR* mutations purchased from Vector Builder (vector IDs available upon request). The following antibodies were used: mouse monoclonal anti-CFTR (ab596, J.R. Riordan, University of North Carolina at Chapel Hill, and Cystic Fibrosis Foundation Therapeutics); mouse monoclonal anti-GAPDH (sc-32233; Biotechnology Inc., Santa Cruz, CA, USA; RRID:AB_627679); and horseradish peroxidase (HRP)-conjugated anti-mouse IgG (ab97023; Abcam; RRID:AB_10679675). The CFTR modulators VX-770, VX-661 and VX-809 were from TargetMol (Wellesley Hills, MA, USA, catalog ID: T2588, T2263, and T2595, respectively), while VX-445 was purchased from MedChemExpress (Monmouth Junction, NJ, USA, catalog ID: HY-111772).

### 2.4. Cell Culture

CFBE41o- and FRT cells stably expressing the halide-sensitive yellow fluorescent protein (HS-YFP) YFP-H148Q/I152L were generated as described previously [27] for the CFBE41o- cells and [28] FRT cells. The CFBE41o- cells were grown in MEM medium (Euroclone, Pero, Milan, Italy) while the FRT cells were cultured in Nutrient Mixture F-12 Ham Coon′s Modification (Merck, Darmstadt, Germania). Both culture mediums were supplemented with 10% FBS, 2 mM L-glutamine, 100 U/mL penicillin, and 100 µg/mL streptomycin. The CFBE41o- or FRT cells were seeded on clear-bottom 96-well black microplates (Corning Life Sciences, Glendale, CA, USA) to perform the YFP-based assays for the CFTR activity. For the CFTR protein expression, the cells were plated on clear-bottom 12-well plates (Eppendorf, Milan, Italy).

### 2.5. Transient Transfection of the Variant CFTR Plasmids

To perform the YFP-based assay, the FRT and CFBE41o- cells expressing the HS-YFP were reverse-transfected onto 96-well plates (Corning Life Sciences) with 0.2 mg per well of the indicated vectors. To evaluate the CFTR expression using the Western blot technique, the cells were reverse-transfected into 12-well plates with 0.8 mg of the indicated vectors. As a transfection reagent, Lipofectamine 2000 (Thermo Fisher Scientific, Waltham, MA, USA) was used. At the time of transfection, the cells were seeded in Opti-MEM™ Reduced Serum Medium (Thermo Fisher Scientific). Six hours post-transfection, the Opti-MEM was replaced with a culture medium without antibiotics. The cells were treated with correctors or the vehicle alone (DMSO) at the indicated concentrations 24 h after transfection and seeding, and then the cells were incubated at 37 °C for an additional 24 h prior to proceeding with the functional HS-YFP-based assay or with the cell lysis.

### 2.6. YFP-Based Assay for the CFTR Activity

The CFTR activity was determined with the HS-YFP microfluorimetric assay [27,28,29,30]. In brief, the cells were washed with PBS (137 mM NaCl, 2.7 mM KCl, 8.1 mM Na_2_HPO_4_, 1.5 mM KH_2_PO_4_, 1 mM CaCl_2_, and 0.5 mM MgCl_2_). To maximally stimulate the CFTR channel, the cells were incubated for 25 min with 60 µL of PBS plus forskolin (fsk; 20 µM) and VX-770 (1 µM). For the CFTR activity determination, the cells were then transferred to a microplate reader (FluoStar Galaxy or Fluostar Optima; BMG Labtech, Offenburg, Germany), equipped with high-quality excitation (HQ500/20X: 500 ± 10 nm) and emission (HQ535/30M: 535 ± 15 nm) filters for YFP (Chroma Technology, Olching, Germany). Each assay consisted of a continuous 14 s YFP fluorescence recording with 2 s before and 12 s after the injection of 165 µL of an iodide-containing solution (PBS with the Cl- replaced with I-; final I- concentration 100 mM). The data were normalized to the initial background-subtracted fluorescence. To determine the I- influx rate, the final 11 s of the data for each well were fitted with an exponential function to extrapolate the initial slope (dF/dt).

### 2.7. CFTR Half-Life Evaluation

The CFBE41o- cells were transiently transfected with the indicated vectors. In order to slow/stop global protein synthesis, 48 h after transfection, the cells were subjected to 24 h cycloheximide chase (CHX; 100 µg/mL) (SigmaAldrich, St. Louis, MO, USA). The cells were then lysed at 0 and 24 h in 1X RIPA buffer and subjected to SDS-PAGE and Western blotting, as described in the dedicated Section 2.8.

### 2.8. Western Blot

After transfection, the FRT or CFBE41o- cells were grown to confluence and treated with correctors into a 12-well plate. On the day of the cell lysis, the cells were washed with ice-cold D-PBS without Ca^2+^/Mg^2+^ and then lysed with RIPA buffer (50 mM Tris-HCl pH 7.4, 150 mM NaCl, 1% Triton X-100, 0.5% Sodium deoxycholate, and 0.1% SDS) containing a complete protease inhibitor cocktail (Roche, Basel, Switzerland). The cell lysates were then processed as follows. Briefly, the lysates were separated by centrifugation at 15,000× *g* for 10 min at 4 °C. A BCA assay (Thermo Fisher Scientific) was used to calculate the supernatant’s protein concentration following the manufacturer’s instructions. A total of 25 mg of total cells were separated onto gradient 4–15% Criterion TGX Precast gels (Bio-rad Laboratories Inc., Hercules, CA, USA), transferred to a nitrocellulose membrane with the Trans-Blot Turbo system (Bio-rad Laboratories Inc.), and analyzed with Western blotting. CFTR and GAPDH were detected using antibodies indicated in the dedicated Section 2.3. Specific bands were visualized with chemiluminescence using the SuperSignal West Dura Substrate (Thermo Fisher Scientific). The Molecular Imager ChemiDoc XRS System (Biorad) was used to monitor the chemiluminescence. ImageJ software (National Institutes of Health) was used to analyze the acquired images. The CFTR bands were analyzed as the region-of-interest (ROI), normalized against the GAPDH loading control.

### 2.9. Statistical Methods

The ordered logistic regression model (OLR), separate for males and females, was applied, using age to predict the clinical grading according to a modified version of the WHO outcome scale [26]. Each patient had a clinical classification equal to: 0 (mild) if the actual patient grading was below the one predicted by the OLR; or 1 (severe) if the grading was above the OLR prediction. The patients with a predicted gradient equal to the actual gradient were excluded from the LASSO analysis, by which we wanted to compare the “extreme ends”. The LASSO logistic regression machine learning approach, applied separately for males and females, was used to extract the relevant genetic features associated with COVID-19 severity and mildness as already described [20]. In this study, we focused on rare (0.1% ≤ MAF < 1%) and ultra-rare (MAF < 0.1%) autosomal recessive (the presence of at least two variants) boolean features on autosomal genes.

The binary association between the presence of the above-identified rare and ultra-rare variants with the WHO outcome scale was assessed using the Fisher exact test.

The Kolmogorov-Smirnov test was used to assess the assumption of the normality of the data distribution. An analysis of variance (ANOVA), followed by a post-hoc test, was used when comparing more than two groups in order to avoid a “multiple comparisons error”. For the normally distributed quantitative variables, a parametric ANOVA was performed.

The statistical significance of the effect of single drug treatments on CFTR activity was tested with a parametric ANOVA followed by the Dunnett multiple comparisons test (all groups against the control group) as a post-hoc test. In the case of combinations of drugs, the statistical significance was verified by an ANOVA, followed by the Tukey test (for multiple comparisons) as a post-hoc test. When comparing the selected pairs of treatments, the statistical significance was tested with an ANOVA, followed by Bonferroni as a post-hoc test.

The normally distributed data were expressed as the mean ± SD, and the significance was two-sided. The differences were considered statistically significant when *p* < 0.05.

## 3. Results

### 3.1. Rare CFTR Variants Contribute to Mildness While Ultra-Rare Variants Contribute to the Severity of COVID-19

The logistic regression model with the LASSO regularization term was used to extract the relevant genes for the classification of severe vs. mild in the cohort of COVID-19-positive patients, adjusted by age and sex. The histogram of the LASSO logistic regression weights (Figure 1) represents the importance of each feature for the classification task. The feature selection was performed separately for the categories of ultra-rare (MAF < 0,1%), rare (0.1% ≤ MAF < 1%), low-frequency (1% ≤ MAF < 5%), and common (MAF ≥ 5%) variants. Furthermore, for each of the above categories, the selections were performed separately if only one variant or two variants were identified. *CFTR* emerged as a feature associated with severity when at least two ultra-rare variants were present (the top diagram in Figure 1) and as a feature associated with mildness when at least two rare variants were present (the bottom diagram in Figure 1).

We then proceeded to identify the nature of the selected features. In hospitalized patients carrying at least two ultra-rare *CFTR* variants, we found 15 different variants. Among them, there were two already known complex alleles (two variants in cis) in patients P2, P9, and P10. The remaining variants were likely in trans (Table 1). We also found two complex alleles (P11 and P12) and two variants in trans (P13) in three not-hospitalized patients. These last three patients were 43, 38, and 21 years old, and all had a negative IPGS value [20]) indicating the presence of other genetic features with a protective effect (Table 2).

Concerning patients with at least two rare variants, all of them (14 females and 27 males) bore the complex allele consisting of cis G574A and R668C variants [G574A;R668C] (Table 3). 

### 3.2. The Rare CFTR [G576A;R558C] Complex Allele Is a Gain-of-Function Allele

In order to understand why the complex allele [G576V;R668C] found in patients described in Table 3 mitigated the COVID-19 outcome, we first characterized the impact of the sequence variations on the CFTR protein. Thus, we investigated the in vitro activity and expression profile of the resulting CFTR protein and compared them to those displayed by the wild-type channel. For this aim, we expressed the variant protein, carrying the complex allele, into two different cell models widely used for studies on CFTR pharmacology and biology, Fischer Rat thyroid (FRT) and immortalized CFBE41o- bronchial epithelial cells, both with a stable expression of the halide-sensitive yellow fluorescent protein (HS-YFP) (Figure 2). The measurements of the CFTR activity using the YFP-based assay revealed a small but significant increase in the halide transport elicited by the cAMP agonist alone or in the presence of ivacaftor to stimulate the CFTR channel fully (Figure 2A).

We then biochemically evaluated the expression of the CFTR protein in FRT and CFBE41o- cells transiently transfected with WT, G576A, R668C, or [G576A;R668C] *CFTR*. As additional controls, we also expressed the F508del mutant (Figure 2B). In the cells transfected with wild-type or [G576A;R668C] *CFTR* the prevalent form was the mature, fully glycosylated CFTR form (band C), while the cells expressing F508del prevailed the immature, core-glycosylated CFTR protein (band B), while the mature form became evident only following 24 h treatment with the combination of the elexacaftor/tezacaftor correctors (Figure 2B). Interestingly, the CFTR band C densitometric analysis highlighted that the expression level of the mature form of [G576A;R668] CFTR was higher than that of the wild-type protein (Figure 2C).

It was demonstrated that the translation rate acts as a substantial contributor to CFTR functional expression [31,32,33]. Indeed, slowing down the rates of translation initiation and elongation enhances WT-CFTR biogenesis (and, therefore, its cell surface expression and function) by improving its folding [31]. The slowing down of the translational rate to improve CFTR expression could be achieved pharmacologically by treating the cells with a submaximal concentration of cycloheximide (CHX) to block protein synthesis [33] partially or by silencing the proteins of the ribosomal stalk [31,32]. Thus, we verified the effect of a translational rate slowdown on WT and [G576A;R668C] *CFTR*. For this aim, we transiently expressed the two CFTR proteins in CFBE41o cells and the following day, we treated the cells with CHX (100 µg/mL) for 24 h. We then compared the expression of the mature CFTR form with SDS-PAGE followed by a Western blotting analysis (Figure 2D,E). The treatment with CHX significantly increased both WT and [G576A;R668C] *CFTR* band C, as also evidenced by the densitometric analysis (Figure 2D,E). Interestingly, the fold-increase in band C obtained over the 24 h CHX treatment was larger for WT CFTR as compared to the variant protein encoded by the complex allele.

### 3.3. Ultra-Rare Simple and Complex Alleles with Reduced CFTR Activity

The characterization of patients reported in Table 1 and Table 2 revealed that they could be either heterozygotes bearing a complex allele (thus, with the variants in cis on the same allele) or compound heterozygotes carrying two *CFTR* variants in trans.

To investigate the impact of the sequence variants on CFTR channel function, we expressed the variant proteins into immortalized CFBE41o- bronchial epithelial cells. We then measured (utilizing the YFP-based assay) the resulting CFTR-mediated halide transport elicited by the cAMP agonist alone or plus ivacaftor in cells treated for 24 h with the vehicle alone (DMSO), the single CFTR corrector Lumacaftor, or the combination of elexacaftor/tezacaftor correctors (Figure 3). Our analyses demonstrated that several sequence variants resulted in decreased CFTR channel activity. In particular, F508del, G542*, and R1066H, three well-known cystic fibrosis-associated mutations, displayed severely reduced activity (with residual activity of up to 20% of the wild-type protein). A similar functional impairment was also previously demonstrated for another variant identified in our cohort, c.1585-1G>A, a severe mutation at the splice acceptor/donor site resulting in little or no CFTR protein (according to “The Clinical and Functional TRanslation of CFTR” (CFTR2) database, available at http://cftr2.org, accessed on 3 November 2022). Interestingly, three of the complex alleles identified in the patients ([A238V;F508del], [F508del;I1027T], and [R74W;V201M;D1270N]) resulted in severely reduced channel activity. In most cases, the variant dysfunction was likely due to protein misfolding causing decreased CFTR processing, trafficking, and activity, as evidenced by the functional rescue observed upon treatment with CFTR-modulating drugs.

Other variants were instead associated with a milder impairment of CFTR function, with a residual activity ranging from 40 to 60% of that of the wild-type protein, as in the case of the K68E and A309D variants. Four variants (A252T, F1052V, I506V, and the complex allele [I506V;D1168G]) caused a modest (but statistically significant) decrease in channel activity (a residual activity of approximately 90% of the wild-type protein). The remaining variants (I328V, R347C, Y625N, V562I, and I807M) were found to display an activity comparable to that of the wild-type protein.

Interestingly, most of the patients in our cohort displayed a global CFTR residual activity (indicating the average activity calculated considering both the *CFTR* alleles) that was between 50 and 91% that of the healthy control.

### 3.4. Association of the CFTR Rare Gain-of-Function Complex Allele and Ultra-Rare Loss-of-Function Alleles with COVID-19 Clinical Outcomes

A cohort of 2585 patients (40.77% females, 59.23% males) was studied (Table 4) after excluding the CF patient (n = 1) and subjects with both gain- and loss-of-function variants (n = 3). In patients younger than 50 years, males were significantly (*p* < 0.01) more prevalent than females in the more severe categories and less prevalent among those not hospitalized (Table 4). Male patients aged ≥ 50 years were more prevalent (*p* < 0.01) among those undergoing invasive mechanical ventilation and less prevalent in the not hospitalized category. Interestingly, in category two, there was a significant prevalence difference in females younger than 50 years vs. older that was not detectable in male individuals (Table 4), indicating a putative protective role of estrogens in female patients.

In the above-described clinical categories, we then studied the distribution of patients carrying either a rare gain-of-function or an ultra-rare loss-of-function *CFTR* variant.

Among the entire cohort, 41 patients (33.63% females and 65.85% males) were identified as carriers of a gain-of-function complex allele (Table 5). Those individuals appeared to be more represented in the milder categories in younger females (age < 50 years) and older males (age ≥ 50 years). However, in females, the most significant prevalence was seen in those aged 50 years or older undergoing CPAP/BiPAP ventilation (Table 5).

In 114 individuals (34.15% females and 66.37% males), we identified ultra-rare loss-of-function variants (Table 6). Overall, they were more represented in the most severe COVID-19 clinical categories; in particular, among those who passed away during their hospitalization (*p* < 0.05 vs. category 4). Interestingly, female patients aged ≥50 years treated with invasive mechanical ventilation represented a significantly higher prevalence of carriers compared to those treated only with low-flow oxygen (*p* < 0.05). Among the males aged 50 years or older who passed away, there was a significantly (*p* < 0.05) higher prevalence of subjects carrying loss-of-function mutations than among males undergoing only low-flow oxygen (Table 6).

### 3.5. COVID-19 as a Mendelian Dominant/Recessive Disorder

Overall, the results described above show that the rare complex [G576V;R668C] allele may mitigate the disease in younger women (age < 50 years) and older men (age ≥ 50 years) via a gain-of-function mechanism with increased activity to 110% (Figure 2 and Table 3). On the contrary, *CFTR* ultra-rare alleles have loss-of-function mechanisms and are associated with severity in both older women and men (age ≥ 50) (Figure 3 and Table 5). Overall, these alleles led to global residual activity of CFTR between 50 to 79%. The reduced function leading to severe COVID-19 could be achieved by having one cis complex allele (dominant disorder) or two hypomorphic alleles in trans (recessive disorder). In the first case, severe COVID-19 would be transmitted as an autosomal dominant disorder, while in the second case, an autosomal recessive disorder.

## 4. Discussion

Here, we demonstrate that the global impairment of CFTR function, either by a strong LOF allele in heterozygosity or compound heterozygosity for two hypomorphic alleles, is associated with COVID-19 severity.

The distribution of the gain-of-function and loss-of-function *CFTR* alleles among the COVID-19 clinical categories showed an age- and sex-dependent pattern. In particular, the gain-of-function complex alleles appeared to be more represented in younger females and older males with mild symptoms, while the loss-of-function ones were significantly more prevalent in older men and women with the most severe disease. This finding may suggest a putative synergic effect of estrogens and *CFTR* variants in modulating the response to SARS-CoV-2 infection. Indeed, estrogens were shown to exert multiple effects on CFTR according to the type of cell studied [34,35]. A naturally occurring estrogen called 17 beta-Estradiol inhibited CFTR activity in vitro, as evidenced by electrophysiological studies on intestinal T84 epithelial monolayers and excised membrane patches [34]. Interestingly, similar effects were obtained with 17 alpha-estradiol, a stereoisomer that fails to bind and activate nuclear estrogen receptors. These findings argued against a genomic-mediated mechanism of action and in favor of a direct interaction of estrogens with the CFTR protein [34]. Besides inhibiting cAMP-mediated chloride secretion, 17 beta-estradiol was also found to increase sodium absorption via the stimulation of PKCδ, EnaC, and Na+/K+ATPase in CF cells [36], leading to further dehydration in the airway’s epithelium surface. On the other hand, in uterine cells, treatment with estrogens increased the expression of CFTR [37]. Thus, we may speculate that *CFTR* gain-of-function variants may either counterbalance the inhibitory effects of estrogens or have a synergic activatory effect, therefore leading to the manifestation of mild COVID-19 disease in younger females and older males, known to have higher estrogen levels compared to post-menopausal women [38]. Further studies are needed to speculate on a possible future therapeutic implication of estrogens’ modulation in CF carrier individuals.

Unexpectedly, the gain-of-function variants were also significantly more prevalent in women aged ≥50 undergoing CPAP/BiPAP ventilation. While a protective hormonal effect in those individuals may not be speculated, a detailed whole exome analysis revealed that other genes could be responsible for the more severe clinical picture. Indeed, in those patients, we identified variants in candidate genes, such as *TNFRSF13B*, *UBA1*, *IRF2BP2*, *FCN3*, *MYOF*, *CFAP54*, *NME8*, and *CFAP46*, involved in the immune system response, inflammatory processes, and respiratory infections. In particular, mutations in *TNFRSF13B*, *UBA1*, *IRF2BP2,* and *FCN3* were already reported to be associated with severe COVID-19 [39,40,41,42]. The *MYOF* (myoferlin) gene modulates *VEGF* signal transduction, which has already been shown to play a key role in life-threatening COVID-19 [43]. *CFAP54*, *NME8,* and *CFAP46* are ciliopathy-related genes that strongly increase the risk of respiratory infections due to the inability to remove inhaled pathogens, including SARS-CoV-2 [44].

Regarding the significant prevalence of loss-of-function *CFTR* alleles/variants in the most severe clinical categories, the finding is in accordance with our previous work [17] and the work of others [45] showing that carriers of single pathogenic *CFTR* variants are more likely to undergo severe COVID-19 with a high risk of 14-day mortality [17]). Estrogen levels and the reported reduced CFTR expression as a consequence of aging [46] may further contribute to the severity of COVID-19 disease in older men and women. Interestingly, it was recently proposed in a preprint that the coronavirus spike protein may bind to CFTR, inhibiting its activity [22] and that the SARS-CoV-2 nucleocapsid protein could interact with Smad3, which downregulated CFTR expression via microRNA-145 [23]. Furthermore, the cleavage sites of the coronavirus 3CL^pro^ proteinase were identified, by computational models, in the intracellular region of CFTR [47]. While these results still need to be confirmed by the scientific community, we could speculate that individuals with a baseline reduction in CFTR function of approximately 50% of the physiological level [48,49] might be more susceptible to further inhibition of CFTR activity via the spike-CFTR binding.

It is noteworthy to mention that in our cohort, there was only one individual affected by CF, a 51-year-old male hospitalized with low-flow oxygen as respiratory support. This is in line with other reports showing that CF patients undergo mild COVID-19 [50]. Indeed, numerous potential protective factors may occur in CF patients: (i) the treatment with CF modulators that reduces the odds of hospitalization [51]; (ii) their accustomization to social distancing and infection control practices; (iii) the impaired membrane expression of CFTR that would not allow for binding with the SARS-CoV-2 spike protein [22]; (iv) their altered signaling of angiotensin converting enzyme (ACE) and ACE2 [52]; (v) the use of certain medications, such as azithromycin, that may protect against infections [50]); and (vi) the systemic levels of ATP intrinsically elevated in CF patients [53,54,55,56,57,58,59]. Based on this, it was hypothesized that ATP supplementation, especially during the first few critical weeks of SARS-CoV-2 infection, may have efficacy in preventing the severe form of COVID-19 [57,58,59].

Further studies are needed to dissect the interaction between CFTR and SARS-CoV-2, including the understanding of how the different SARS-CoV-2 variants may affect the clinical presentation of COVID-19 in individuals with loss-of-function CFTR genotypes. In conclusion, our work highlights the importance of the CFTR genetic profile in determining COVID-19 presentation and progression, opening new perspectives in personalized approaches for COVID-19.

## Figures and Tables

**Figure 1 cells-11-04096-f001:**
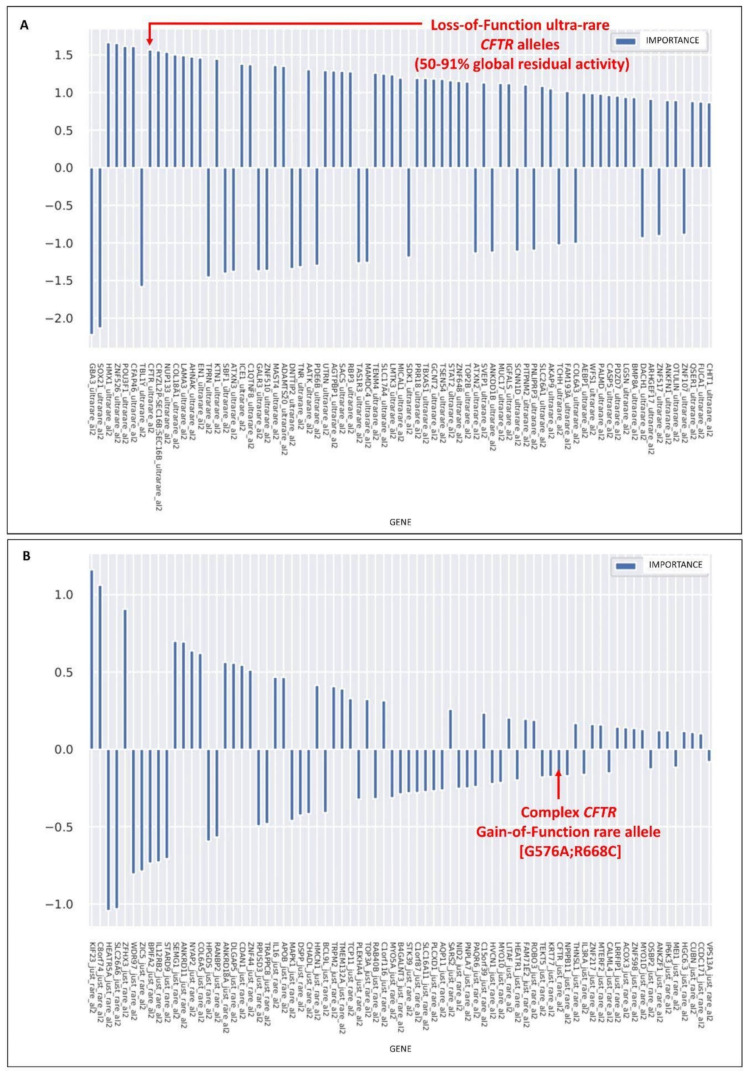
*CFTR* alleles associated with mild or severe COVID-19. LASSO logistic regression analysis on the boolean representation of ultra-rare variants (the presence of at least two variants, al2), (**A**) and rare variants (the presence of at least two variants, al2), (**B**) of all autosomal genes (see Fallerini et al. 2022 for complete representations). The upward histograms (positive weights) represent the features associated with severe COVID-19, whereas the downward histograms (negative weights) represent the features associated with mild COVID-19. (**A**) The presence of at least two, in cis or trans, *CFTR* ultra-rare loss-of-function variants was picked up as one of the most important features associated with COVID-19 severity. The loss-of-function variants were defined on the basis of a residual global *CFTR* activity reduction between 50–79%. (**B**). The gain-of-function *CFTR* complex allele [G576V;R668C] was picked up as one of the most important features associated with COVID-19 mildness.

**Figure 2 cells-11-04096-f002:**
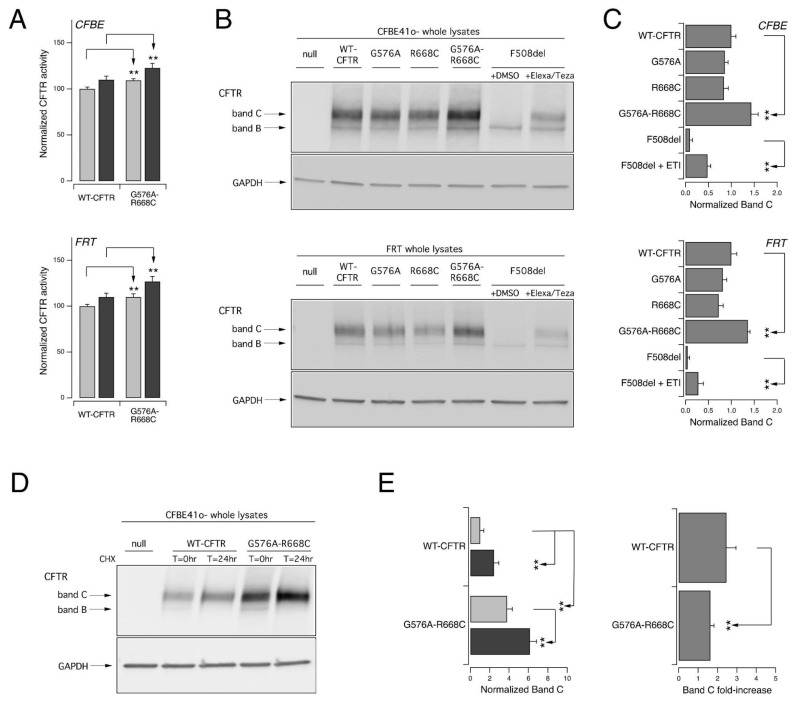
The complex allele [G576V;R668C] increases the activation and maturation of the CFTR channel. The functional and biochemical analysis of the [G576A;R668C] *CFTR* complex allele on heterologous expression systems. (**A**). Increase in CFTR activation. The bar graphs show the activity of [G576A;R668C] *CFTR* and, for comparison, WT-CFTR transiently expressed in CFBE41o- (upper panel) or FRT (lower panel) cells stably expressing HS-YFP. The CFTR activity was determined as a function of the YFP quenching rate following the iodide influx in cells stimulated with FSK alone (20 µM; light gray) or with FSK plus ivacaftor (1 µM; dark gray). The data are means ± SD (n = 3). The asterisks indicate statistical significance vs. the WT-CFTR protein: **, *p* < 0.01. (**B**). Increase in CFTR maturation. Representative Western blot images showing the electrophoretic mobility of [G576A;R668C] and, for comparison, wild-type, G576A, R668C, and F508del CFTR transiently expressed in CFBE41o- (upper panel) or FRT (lower panel) cells. The whole lysates derived from cells not expressing CFTR (null cells) are shown as controls for antibody specificity. In the case of F508del, the cells were treated for 24 h with DMSO alone (vehicle) or Elexa/Teza (3 µM/10 µM) to correct the mutant misfolding. The arrows indicate the complex-glycosylated (band C) and core-glycosylated (band B) forms of the CFTR protein. (**C**). CFTR band C densitometry of the Western blot experiments. The data are means ± SD (n = 3). The asterisks indicate statistical significance vs. WT-CFTR protein: **, *p* < 0.01. (**D**). CFTR half-life evaluation. Representative Western blot image showing the CFTR expression pattern of [G576A;R668C] CFTR and, for comparison, WT-CFTR transiently expressed in CFBE41o- cells at different time points (T = 0 and 24 h) following the CHX-induced block of protein synthesis. The whole lysates derived from cells not expressing CFTR (null cells) are shown as controls for antibody specificity. (**E**). The quantification of the normalized CFTR band C (left graph) following CHX treatment (T = 0 h, light gray; T = 24 h, dark gray) and band C fold-increase (right graph) over 24 h CHX treatment, obtained from experiments detailed in D, normalized with the initial value of band C. The data are means ± SD (n = 3). The asterisks indicate statistical significance vs. WT-CFTR protein: **, *p* < 0.01.

**Figure 3 cells-11-04096-f003:**
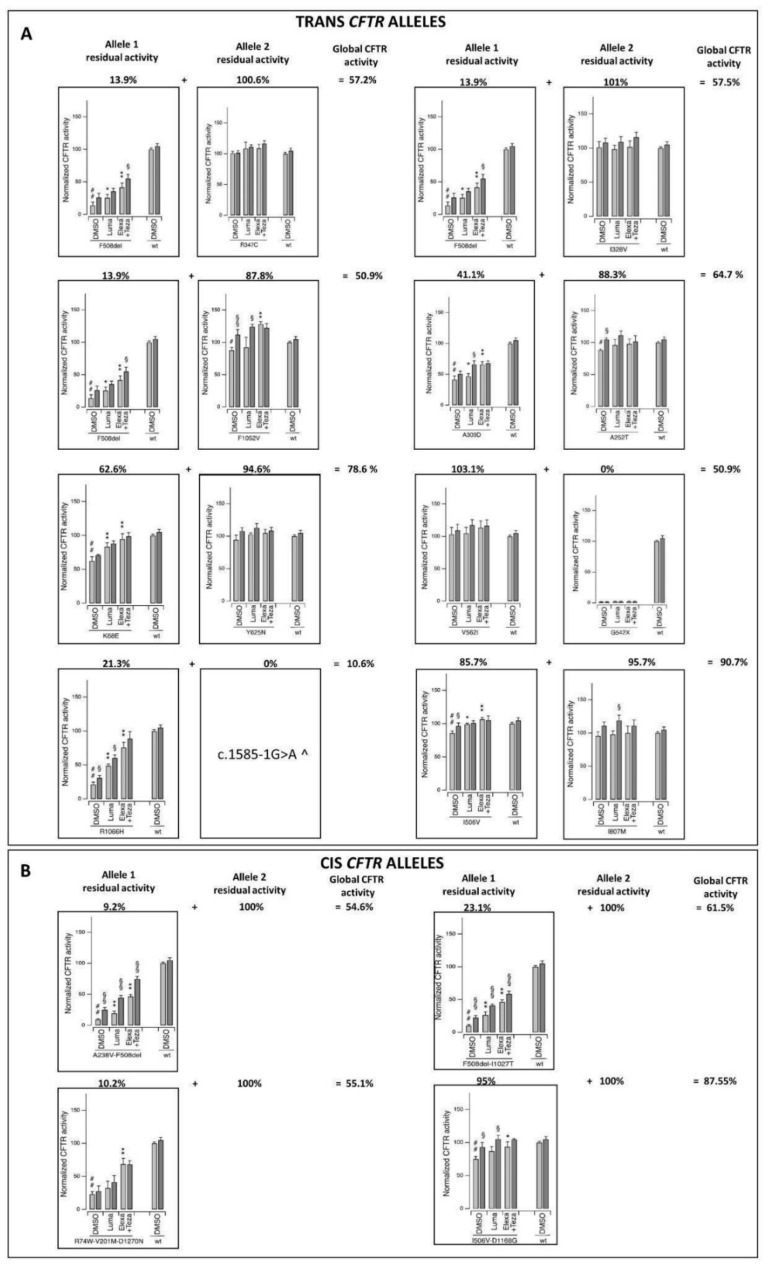
Functional analysis of *CFTR* ultra-rare variants on heterologous expression systems identified loss-of-function alleles with reduced activity. (**A**) Functional analysis of trans CFTR alleles. (**B**) Functional analysis of cis CFTR alleles. The bar graphs show the activity of the different *CFTR* variants under investigation and, for comparison, WT-*CFTR* transiently expressed in CFBE41o- cells stably expressing HS-YFP. CFTR activity was determined as a function of the YFP quenching rate following the iodide influx in the cells stimulated with FSK alone (20 µM; light gray) or with FSK plus ivacaftor (1 µM; dark gray). The data are means ± SD (n = 3). The symbols indicate the statistical significance: # *p* < 0.05 vs. WT-CFTR protein; ## *p* < 0.01 vs. WT-CFTR protein; * *p* < 0.05, ** *p* < 0.01 vs. DMSO-treated, FSK-stimulated variant protein; §, *p* < 0.05, §§, *p* < 0.01 vs. FSK-stimulated variant protein (upon the same chronic treatment). ^ this type of variant is expected to result in little or no CFTR protein (see CFTR2 database at https://cftr2.org, accessed on 3 November 2022).

**Table 1 cells-11-04096-t001:** *CFTR* Loss-of-Function (LOF) alleles in hospitalized patients.

Nucleotide Change	Amino Acid Change	dbSNP	CADD	gnomAD_ NFE	Cis/ Trans	Alleles’ Residual Activity (%)	Global Residual Activity (%)	N. of Patients	Clinical Category	Age	Sex	Long COVID (Yes/No)	Patient ID	IPGS
c.1039C>T	p.R347C	rs39750814	34	0.0000088	trans	100.6	57.2	1	2	32	M	Yes	P1	180
c.1520_1522del	p.F508del	rs113993960	n.a.	0.012		13.9								
c.[713C>T;1520_1522del]	p.[A238V; F508del]	rs39750878; rs113993960	26; n.a.	n.a; 0.012	cis	9.2	54.6	1	2	30	M	Yes	P2	250
WT	WT	n.a.	n.a.	n.a.		100 ^								
c.3154T>G	p.F1052V	rs15021278	31	0.009	trans	87.8	50.9	1	4	52	M	Yes	P3	−80
c.1520_1522del	p.F508del	rs113993960	n.a.	0.012		13.9								
c.202A>G	p.K68E	rs39750833	24	0.0001	trans	62.6	78.6	1	4	42	M	n.a.	P4	32
c.1873T>A	p.Y625N	rs760390633	24	0.000018		94.6								
c.A982A>G	p.I328V	n.a.	29	n.a.	trans	101	57.5	1	3	63	M	Yes	P5	215
c.1520_1522del	p.F508del	rs113993960	n.a.	0.012		13.9								
215c.754G>A	p.A252T	n.a.	24	n.a.	trans	88.3	64.7	1	4	91	F	Yes	P6	−174
c.926C>A	p.A309D	n.a.	32	n.a.		41.1								
c.1624G>T	p.G542 *	rs11399395	39	0.0004	trans	0	51.55	1	3	77	F	n.a.	P7 °	176
c.1684G>A	p.V562I	rs1800097	24	0.0002		103.1								
c.3197G>A	p.R1066H	rs121909019	34	0.000076	trans	21.3	10.6	1	4	51	M	Yes	P8	−15
c.1585-1G>A	n.a.	rs76713772	26	0.0001		0								
c.[1516A>G;3503A>G]	p.[I506V;D1168G]	rs1800091; rs150326506	24; 28	0.0005; na	cis	95	87.55	2	1	65	M	No	P9	243
WT	WT	n.a.	n.a.	n.a.		100 ^			2	69	M	n.a.	P10	276

° Patient P7 was deceased upon follow-up. * Complex allele 1 = c.[220>T; 3808G>A; c.601G>A], p.[R74W;D1270N; V201M]. ^ totally conserved activity of the wild-type allele. Patient P8 had a clinical diagnosis of cystic fibrosis.

**Table 2 cells-11-04096-t002:** *CFTR* Loss-of-Function (LOF) alleles in not-hospitalized patients.

Nucleotide Change	Amino Acid Change	dbSNP	CADD	gnomAD_ NFE	Cis/ Trans	Residual Allele Activity (%)	Global Residual Activity (%)	N. of Patients	Clinical Category	Age	Sex	Long-COVID (Yes/No)	Patient ID	IPGS
Complex Allele 1 *	Complex Allele 1 *	rs115545701; rs11971167; rs138338446	24; 34; 27	0.0003; 0.0001; 0.0002	cis	23.1	61.6	1	6	38	M	NA	P11	−170
WT	WT	n.a.	n.a.	n.a.		100 ^								
Complex allele 2 ^#^	Complex allele 2 ^#^	rs1800112; rs113993960	17; na	0.003; 0.0123	cis	10.2	55.1	1	6	21	F	Yes	P12	−22
WT	WT	n.a.	n.a.	n.a		100 ^								
c.1516A>G	p.I506V	rs1800091	24	0.0005	trans	85.7	90.7	1	6	43	F	NA	P13	−32
c.2421A>G	p.I807M	rs1800103	22	0.0006		95.7								

* Complex allele 1 = c.[220>T; 3808G>A; c.601G>A], p.[R74W;D1270N; V201M]. ^#^ Complex allele 2 = c.[3080T>C; c.1520_1522del], p.[I1027T; F508del]. CADD, combined annotation dependent depletion; GnomAD, genome aggregation database; NFE, non-Finnish European; residual allele activity: percentage of the residual activity of the single alleles; Global residual activity: percentage of the residual activity of the protein; Clinical category: 1, death; 2, hospitalized, receiving invasive mechanical ventilation; 3, hospitalized, receiving continuous positive airway pressure (CPAP) or bilevel positive airway pressure (BiPAP) ventilation; 4, hospitalized, receiving low-flow supplemental oxygen; 5, hospitalized, not receiving supplemental oxygen; 6, not hospitalized. IPGS: integrated polygenic score for severity prediction ([20] Fallerini et al. 2022) ^ totally conserved activity of the wild-type allele.

**Table 3 cells-11-04096-t003:** *CFTR* Gain-of-Function (GOF) complex allele [G574A;R668C] in all patients.

Clinical Category	Females (n.)	Median Age (IQR)	Total n. Patients
Cat. 1	F (1)	84 y	1
Cat. 2	F (1)	57 y	2
	M (1)	75 y	
Cat. 3	6(F)	81.5 y (69.3–85.5)	15
	9 (M)	71 y (56.3–76.5)	
Cat. 4	2 (F)	77;82 y	11
	9(M)	62 y (54–72.5)	
Cat. 5	2 (F)	43;53 y	4
	2(M)	57;64 y	
Cat. 6	2 (F)	44;45 y	8
	6 (M)	57.5 y (42.5–63.5)	

Clinical category: 1, death; 2, hospitalized, receiving invasive mechanical ventilation; 3, hospitalized, receiving continuous positive airway pressure (CPAP) or bilevel positive airway pressure (BiPAP) ventilation; 4, hospitalized, receiving low-flow supplemental oxygen; 5, hospitalized, not receiving supplemental oxygen; 6, not hospitalized.

**Table 4 cells-11-04096-t004:** Prevalence of male and female patients in the COVID-19 clinical categories according to age.

		Female		Male	
	Total (n = 2585)	Age < 50 y (n = 289)	Age ≥ 50 y (n = 765)	Age < 50 y (n = 317)	Age ≥ 50 y (n = 1214)
COVID Outcome Scale, No. (%)					
Cat. 1 (Death)	164 (6.34)	2 (0.69)	68 (8.89) **	1 (0.32)	93 (7.66) **
Cat. 2 (Invasive mechanical ventilation)	180 (6.96)	6 (2.08)	36 (4.71) *	23 (7.26) °	115 (9.47) °
Cat. 3 (CPAP/BiPAP)	584 (22.59)	24 (8.30)	163 (21.31) **	55 (17.35) °	342 (28.17) **
Cat. 4 (Hospitalized, receiving supplemental low-flow oxygen)	850 (32.88)	37 (12.8)	291 (38.04) **	79 (24.92) °	443 (36.49) **
Cat. 5 (Hospitalized, not receiving supplemental oxygen)	347 (13.42)	58 (20.07)	96 (12.55) *	63 (19.87)	130 (10.71) **
Cat. 6 (Not hospitalized)	460 (17.79)	162 (56.6)	111 (14.49) **	96 (30.28) °	91 (7.5) **,°

* *p* < 0.05 vs. female age < 50 with Fisher exact test. ** *p* < 0.001 vs. age < 50 same-sex with Fisher exact test. ° *p* < 0.001 vs. same age female with Fisher exact test. The percentage reported in the first column refers to the whole cohort (n = 2586). The one reported in the other columns refers to the group of patients with their age and sex reported in the corresponding column.

**Table 5 cells-11-04096-t005:** Patients with the *CFTR* gain-of-function complex allele [G576V;R668C].

		Female		Male	
	Total	Age < 50 y (n = 289)	Age ≥ 50 y (n = 765)	Age < 50 y (n = 317)	Age ≥ 50 y (n = 1214)
Individuals	41	3 (1.04)	11 (1.44)	4 (1.26)	23 (1.89)
COVID Outcome Scale, No. (%)					
Cat. 1 (Death; n = 164)	1 (0.61)	0	1 (1.47)	0	0
Cat. 2 (Invasive mechanical ventilation; n = 180)	2 (1.11)	0	1 (2.78)	0	1 (0.87)
Cat. 3 (CPAP/BiPAP; n = 584)	15 (2.57)	0	6 (3.68) °	1 (1.82)	8 (2.34)
Cat. 4 (Hospitalized, receiving supplemental low-flow oxygen; n = 850)	11 (1.29)	0	2 (0.69)	1 (1.27)	8 (1.81)
Cat. 5 (Hospitalized, not receiving supplemental oxygen; n = 347)	4 (1.15)	1 (1.72)	1 (1.04)	0	2 (1.54)
Cat. 6 (Not hospitalized; n = 460)	8 (1.74)	2 (1.23)	0	2 (2.08)	4 (4.4) *

° *p* < 0.05 vs. category 4 same age and sex with Fisher exact. * *p* = 0.06 vs. category 1 same age and sex with Fisher exact. The percentage reported in the first column refers to the number of patients in the corresponding clinical category. The one reported in the other columns refers to the group of patients with their age and sex reported in the corresponding column and their clinical category in the related raw data.

**Table 6 cells-11-04096-t006:** Patients with *CFTR* ultra-rare alleles with reduced residual activity.

		Female		Male	
	Total (n = 2585)	Age < 50 y (n = 289)	Age ≥ 50 y (n = 765)	Age < 50 y (n = 317)	Age ≥ 50 y (n = 1214)
Individuals	114	(n = 13)	(n = 25)	(n = 20)	(n = 56)
COVID Outcome Scale, No. (%)					
Cat. 1 (Death; n = 164)	13 (7.93) *,°	0	4 (5.88)	0	9 (9.68) *
Cat. 2 (Invasive mechanical ventilation; n = 180)	10 (5.56)	0	3 (8.33) *	3 (13.04)	4 (3.48)
Cat. 3 (CPAP/BiPAP; n = 584)	27 (4.62)	1 (4.17)	7 (4.29)	3 (5.45)	16 (4.68)
Cat. 4 (Hospitalized, receiving supplemental low-flow oxygen; n = 850)	29 (3.41)	2 (5.41)	4 (1.37)	5 (6.33)	18 (4.06)
Cat. 5 (Hospitalized, not receiving supplemental oxygen; n = 347)	13 (3.75)	2 (3.45)	3 (3.12)	2 (3.17)	6 (4.62)
Cat. 6 (Not hospitalized; n = 460)	22 (4.78)	8 (4.94)	4 (3.6)	7 (7.29)	3 (3.30)

* *p* < 0.05 vs. category 4. ° *p* = 0.05 vs. category 5. The percentage reported in the first column refers to the number of patients in the corresponding clinical category. The one reported in the other columns refers to the group of patients with their age and sex reported in the corresponding column and their clinical category in the related raw data.

## Data Availability

The data and samples referenced here are housed in the GEN-COVID patient registry and the GEN-COVID biobank and are available for consultation. You may contact Alessandra Renieri (e-mail: alessandra.renieri@unisi.it).

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
