# Peer review of "Gain- and Loss-of-Function CFTR Alleles Are Associated with COVID-19 Clinical Outcomes"

_cells, 2022, doi:10.3390/cells11244096_

Round 1

Reviewer 1 Report

The authors  has previously shown that CF carriers with a single pathogenic variant are more likely to undergo severe COVID with higher risk of 14 day mortality. They subsequently reported that COVID-19 spread and fatality in 37 countries corresponded to the geographical distribution of carrier frequency. Here they have used a new Machine Learning post-Mendelian model on a 2,885 member cohort CF patients and their  COVID-19 outcomes from among COVID-19 patients enrolled within the GEM-COVID Multicenter Study,  in which the CFTR gene was an important factor in modulating COVID-19 outcome.  The hypothesis was that  the overall activity of CFTR, determined mainly by genetic variants that each individual varies, would be a prognostic  factor influencing COVID-19 severity. To test this hypothesis they analyzed all CFTR variants, and all other detectable genes by Whole Genome Sequencing (WGS),   and compared heterozygous variants with COVID-19 outcomes. Then they analyzed the  the same variants in vitro in two cell types:  FRT cells and CFBE41-o- cells.  The outcome,  briefly, was that the relationship between COVID mildness and severity in patients paralleled the range of CFTR activity from 110% to 50%.  In the several instances where a gain-in-function was noted for in vitro analysis, but poor COVID-19 outcome, the basis of the discordance was traced to mutations in bystander genes associated with poor CF patient function and poor COVID-19 outcome.

Strengths:

1. The hypothesis was based on  previous findings showing that (i) carriers with heterozygous CFTR mutations did poorly insofar as COVID-19 outcome; that (ii) carrier frequency was corelated geographically  with COVID-19 outcome; and (iii) evidence that SARS-CoV-2 virus and certain viral proteins caused reduction in CFTR expression.

2. The new study was based on a large number of patients, all of whom had been analyzed by WGS.

3.  The investigators used a new Machine Learning post-Mendelian model to test the hypothesis in patients.

4. The investigators tested the epidemiological data on parallel functional tests in not one but two classical cell models for measuring CFTR function. They found a relationship between COVID mildness and severity in patients that mostly paralleled the range of CFTR activity. Thus cell specific effects seemed to be excluded.

5. Finally, when confronted with a few CF patients with in vitro gain-in-function mutations, but poor COVID-19 outcomes, the investigators identified bystander genes that were previously identified as causing poor CF outcomes or poor COVID-19 outcomes.

6. Methods and Statistical tests are appropriate and correctly applied.

Weaknesses/Concerns:  

1. No major weaknesses in this outstanding paper.

2. Minor corrections (The  authors should review the reference numbers; I did not test all of them. )

2.1. Line 15 on page 2: The nucleocapsid reference is #24, not $23.

2.22. 6th line from the end of the manuscript, in the  discussion. Reference #22 should

         be Reference #23.

Author Response

Response to Reviewer 1 Comment

Weaknesses/Concerns:  

  1. No major weaknesses in this outstanding paper.

We are very grateful for the positive feedback on our research article.

  1. Minor corrections (The authors should review the reference numbers; I did not test all of them. )

As requested, we have reviewed all the reference numbers and we made some corrections.  

2.1. Line 15 on page 2: The nucleocapsid reference is #24, not $23.

The reference number is now correct. 

2.22. 6th line from the end of the manuscript, in the discussion. Reference #22 should be Reference #23.

We have checked this reference and in our opinion it is correct.

Reviewer 2 Report

The article is an impressive testament to the power of CF genetic analysis.

Tables 2 to 4 contain important results of the study and would benefit

from some further discussion and consideration.

The phenotypic basis for the genotypic analysis is crucial in terms of 

clinical therapeutic benefit of the study.  

The pathophysiolical implications of the study are most important 

in terms extending the important findings. 

1) It would be important to further discuss and define the genetic drift

of the COVID virus and how these changes might affect the disease severity

in the CF populations. 

2) The discussion of estrogenic implications of the genetic findings is interesting but it would be helpful to discuss how this might lead to therapeutic interventions.

 However, one area that is missing from the paper is any mention of the role of purines in CF and the roles of the CFTR. 

The airway in CF has been shown to be have depleted ATP and adenosine levels. These levels are difficult to measure. 

Blood and system levels are elevated in CF knockout mice and patients with 

DF508 mutations.

Early CF covid patients were most DF508 and showed improved survival compared to the general population.

Several factors come to play. Decreasing blood ATP levels with CF and non CF population aging. Increasing age leads to worsening COVID outcome.

It is possible to increase blood ATP by iv ATP infusion or oral ATP administration and survival of COVID infection improves with increasing blood ATP (RBC and plasma levels). 

Airway ATP and adenosine levels are relatively depleted in CF due to decreased CFTR mediated Panexin ATP release to airway apical surface. 

Aerosolized adenosine has been demonstrated by several groups in Italy to improve COVID survival for ventilator bound COVID patients. This therapy might be considered in the worst CF patient based on the genetic analysis. 

It would be most interesting to measure blood /plasma ATP levels in the various CF genotypes and correlate with COVID response to begin to get a handle on the underlying pathophysiology of genetics and CF pathophysiology and the interaction with Covid. The combined approach would give insight to COVID CF interaction and would extend the underating of CF.

Relevant references that I believe might be included in the paper discussion:  

Purinergic Signal  

2021 Sep;17(3):399-410.  doi: 10.1007/s11302-021-09771-0. Epub 2021 May 10.

Cystic fibrosis improves COVID-19 survival and provides clues for treatment of SARS-CoV-2

Edward H Abraham 1 2Guido Guidotti 3Eliezer Rapaport 4David Bower 5Jack Brown 6Robert J Griffin 7Andrew Donnelly 8Ellen D Waitzkin 9Kenon Qamar 10Mark A Thompson 10Sukumar Ethirajan 10Kent Robinson 11     iScience Article Pannexin-1 channel opening is critical for COVID-19 pathogenesis Ross Luu,1 Silvana Valdebenito,1 Eliana Scemes,2 Antonio Cibelli,3 David C. Spray,3 Maximiliano Rovegno,4 Juan Tichauer,4 Andrea Cottignies-Calamarte,5,6 Arielle Rosenberg,5,6,7 Calude Capron,8 Sandrine Belouzard,9 Jean Dubuisson,9 Djillali Annane,10,11 Geoffroy Lorin de la Grandmaison,12 Elisabeth Cramer-Borde´ , 13 Morgane Bomsel,14,15 and Eliseo Eugenin1,16, * 2000 Sep;20(5):348-53.  doi: 10.1046/j.1365-2281.2000.00272.x.

Increased circulating levels of plasma ATP in cystic fibrosis patients

A S Lader 1A G PratG R Jackson JrK L ChervinskyA LapeyT B KinaneH F Cantiello      

  • Published: 01 May 1996

Cystic fibrosis hetero–and homozygosity is associated with inhibition of breast cancer growth

  • 190 Accesses

  • 50 Citations

  • Altmetric

  • Metrics

                J Physiol. 2010 Dec 1; 588(Pt 23): 4605–4606.   doi: 10.1113/jphysiol.2010.200113 PMCID: PMC3010124 PMID: 21123201

CFTR channels and adenosine triphosphate release: the impossible rendez-vous revisited in skeletal muscle

Frédéric Becq

Front Immunol. 2021; 12: 613070.  Published online 2021 Mar 18. doi: 10.3389/fimmu.2021.613070 PMCID: PMC8012541 PMID: 33815368

Efficacy and Effect of Inhaled Adenosine Treatment in Hospitalized COVID-19 Patients

Massimo Caracciolo,1, Pierpaolo Correale,2, Carmelo Mangano,3, Giuseppe Foti,3 Carmela Falcone,4 Sebastiano Macheda,5 Maria Cuzzola,6 Marco Conte,6 Antonella Consuelo Falzea,2 Eleonora Iuliano,2 Antonella Morabito,7 Michele Caraglia,8,9 Nicola Polimeni,5 Anna Ferrarelli,4 Demetrio Labate,5 Marco Tescione,5 Laura Di Renzo,10 Gaetano Chiricolo,11 Lorenzo Romano,12,* and Antonino De Lorenzo10   PLISone

RESEARCH ARTICLE

Therapeutic effects of adenosine in high flow 21% oxygen aereosol in patients with Covid19-pneumonia

  • Pierpaolo Correale,
  • Massimo Caracciolo,
  • Federico Bilotta ,
  • Marco Conte,
  • Maria Cuzzola,
  • Carmela Falcone,
  • Carmelo Mangano,
  • Antonella Consuelo Falzea,
  • Eleonora Iuliano,
  • Antonella Morabito,
  • Giuseppe Foti,
  • Antonio Armentano,
  • Michele Caraglia,
  •  [ ... ],
  • Sebastiano Macheda
  • [ view all ]

  •  
  •  
  •  
  •  
  •  
  •  
  •  

  • Published: October 8, 2020
  • https://doi.org/10.1371/journal.pone.0239692
search menu     Journals    JCM    Volume 9    Issue 9    10.3390/jcm9093045    settings   Review

Can Adenosine Fight COVID-19 Acute Respiratory Distress Syndrome?

by  Carmela Falcone  1,†, Massimo Caracciolo  2,†, Pierpaolo Correale  3, Sebastiano Macheda  4, Eugenio Giuseppe Vadalà  4, Stefano La Scala  4, Marco Tescione  4, Roberta Danieli  5, Anna Ferrarelli  1, Maria Grazia Tarsitano  6, Lorenzo Romano  7,* and Antonino De Lorenzo  8     1 Unit of Radiology, Grande Ospedale Metropolitano Bianchi Melacrino Morelli, 89124 Reggio Calabria, Italy 2 Unit of Intensive Postoperative Therapy, Grande Ospedale Metropolitano Bianchi Melacrino Morelli, 89124 Reggio Calabria, Italy 3 Medical Oncology Unit, Grande Ospedale Metropolitano Bianchi Melacrino Morelli, 89124 Reggio Calabria, Italy 4 Unit of Intensive Care Medicine and Anesthesia, Grande Ospedale Metropolitano Bianchi Melacrino Morelli, 89124 Reggio Calabria, Italy 5 Department of Human Sciences and Promotion of the Quality of Life, University San Raffaele, 00166 Rome, Italy 6 Department of Experimental Medicine, University of Rome Sapienza, 00161 Rome, Italy 7 School of Specialization in Food Science, University of Rome Tor Vergata, 00133 Rome, Italy 8 Section of Clinical Nutrition and Nutrigenomics, Department of Biomedicine and Prevention, University of Rome Tor Vergata, 00133 Rome, Italy * Author to whom correspondence should be addressed. These authors share equal contributions. J. Clin. Med. 20209(9), 3045; https://doi.org/10.3390/jcm9093045 Received: 31 July 2020 / Revised: 16 September 2020 / Accepted: 16 September 2020 /   

Can Adenosine Fight COVID-19 Acute Respiratory Distress Syndrome?

by  Carmela Falcone  1,†, Massimo Caracciolo  2,†, Pierpaolo Correale  3, Sebastiano Macheda  4, Eugenio Giuseppe Vadalà  4, Stefano La Scala  4, Marco Tescione  4, Roberta Danieli  5, Anna Ferrarelli  1, Maria Grazia Tarsitano  6, Lorenzo Romano  7,* and Antonino De Lorenzo  8     1 Unit of Radiology, Grande Ospedale Metropolitano Bianchi Melacrino Morelli, 89124 Reggio Calabria, Italy 2 Unit of Intensive Postoperative Therapy, Grande Ospedale Metropolitano Bianchi Melacrino Morelli, 89124 Reggio Calabria, Italy 3 Medical Oncology Unit, Grande Ospedale Metropolitano Bianchi Melacrino Morelli, 89124 Reggio Calabria, Italy 4 Unit of Intensive Care Medicine and Anesthesia, Grande Ospedale Metropolitano Bianchi Melacrino Morelli, 89124 Reggio Calabria, Italy 5 Department of Human Sciences and Promotion of the Quality of Life, University San Raffaele, 00166 Rome, Italy 6 Department of Experimental Medicine, University of Rome Sapienza, 00161 Rome, Italy 7 School of Specialization in Food Science, University of Rome Tor Vergata, 00133 Rome, Italy 8 Section of Clinical Nutrition and Nutrigenomics, Department of Biomedicine and Prevention, University of Rome Tor Vergata, 00133 Rome, Italy * Author to whom correspondence should be addressed. These authors share equal contributions. J. Clin. Med. 20209(9), 3045; https://doi.org/10.3390/jcm9093045Published: 21 September 2020 (This article belongs to the Special Issue COVID-19: Diagnostic Imaging and Beyond - Part I) Download PDF  Browse Figures   Citation Export

Abstract

Coronavirus disease

Author Response

Response to Reviewer 2 Comments

Comments and Suggestions for Authors

The article is an impressive testament to the power of CF genetic analysis. Tables 2 to 4 contain important results of the study and would benefit from some further discussion and consideration. The phenotypic basis for the genotypic analysis is crucial in terms of clinical therapeutic benefit of the study. The pathophysiolical implications of the study are most important in terms extending the important findings. 

1) It would be important to further discuss and define the genetic drift of the COVID virus and how these changes might affect the disease severity in the CF populations. 

We thank the Reviewer for this suggestion. Although we have no specific data about the virus variant for our study population, we can assume that the SARS-CoV-2 infections were due to the alpha (B.1.1.7) and delta (B.1.617.2) variants due to their prevalence in Italy during the study period. We have specified this aspect in the Material and Methods section (Patients) on page 4 of the manuscript. We also added this concept in the Discussion section on page 25.

2) The discussion of estrogenic implications of the genetic findings is interesting but it would be helpful to discuss how this might lead to therapeutic interventions.  

We agree with the Reviewer on further discussing the therapeutic implications  of our results. Further studies are needed to speculate on the safety and efficacy of estrogens' modulation in COVID-19 patients. We added a comment about this in the Discussion section on page 24.

However, one area that is missing from the paper is any mention of the role of purines in CF and the roles of the CFTR.

The airway in CF has been shown to be have depleted ATP and adenosine levels. These levels are difficult to measure. Blood and system levels are elevated in CF knockout mice and patients with DF508 mutations.

Early CF covid patients were most DF508 and showed improved survival compared to the general population.

Several factors come to play. Decreasing blood ATP levels with CF and non CF population aging. Increasing age leads to worsening COVID outcome.

It is possible to increase blood ATP by iv ATP infusion or oral ATP administration and survival of COVID infection improves with increasing blood ATP (RBC and plasma levels). 

Airway ATP and adenosine levels are relatively depleted in CF due to decreased CFTR mediated Panexin ATP release to airway apical surface. 

Aerosolized adenosine has been demonstrated by several groups in Italy to improve COVID survival for ventilator bound COVID patients. This therapy might be considered in the worst CF patient based on the genetic analysis. 

We thank the Reviewer for this suggestion. We agree  that several other factors could be implicated in the severity of CF patients and in the role of CFTR variant. We added a comment about this in the Discussion section on page 25.

It would be most interesting to measure blood /plasma ATP levels in the various CF genotypes and correlate with COVID response to begin to get a handle on the underlying pathophysiology of genetics and CF pathophysiology and the interaction with Covid. The combined approach would give insight to COVID CF interaction and would extend the underating of CF.

We thank the Reviewer for this suggestion and we agree that this could be another approach to dissect the CF/COVID-19 axis. Unfortunately, we have no biological samples needed to perform this measurement, but we will consider it  in further future studies. 

Relevant references that I believe might be included in the paper discussion:  

Abraham, Edward H et al. “Cystic fibrosis improves COVID-19 survival and provides clues for treatment of SARS-CoV-2.” Purinergic signalling vol. 17,3 (2021): 399-410. doi:10.1007/s11302-021-09771-0

Lader, A S et al. “Increased circulating levels of plasma ATP in cystic fibrosis patients.” Clinical physiology (Oxford, England) vol. 20,5 (2000): 348-53. doi:10.1046/j.1365-2281.2000.00272.x

Abraham, E H et al. “Cystic fibrosis hetero- and homozygosity is associated with inhibition of breast cancer growth.” Nature medicine vol. 2,5 (1996): 593-6. doi:10.1038/nm0596-593

Becq, Frédéric. “CFTR channels and adenosine triphosphate release: the impossible rendez-vous revisited in skeletal muscle.” The Journal of physiology vol. 588,Pt 23 (2010): 4605-6. doi:10.1113/jphysiol.2010.200113

Caracciolo, Massimo et al. “Efficacy and Effect of Inhaled Adenosine Treatment in Hospitalized COVID-19 Patients.” Frontiers in immunology vol. 12 613070. 18 Mar. 2021, doi:10.3389/fimmu.2021.613070

Correale, Pierpaolo et al. “Therapeutic effects of adenosine in high flow 21% oxygen aereosol in patients with Covid19-pneumonia.” PloS one vol. 15,10 e0239692. 8 Oct. 2020, doi:10.1371/journal.pone.0239692

Falcone, Carmela et al. “Can Adenosine Fight COVID-19 Acute Respiratory Distress Syndrome?.” Journal of clinical medicine vol. 9,9 3045. 21 Sep. 2020, doi:10.3390/jcm9093045

We thank the Reviewer for this suggestion and we added all the suggested references in the manuscript.